# Reduction of Vision-Based Models for Fall Detection

**DOI:** 10.3390/s24227256

**Published:** 2024-11-13

**Authors:** Asier Garmendia-Orbegozo, Miguel Angel Anton, Jose David Nuñez-Gonzalez

**Affiliations:** 1Fundación Tecnalia Research & Innovation, Basque Research and Technology Alliance (BRTA), 20009 San Sebastian, Spain; mangel.anton@tecnalia.com; 2Department of Applied Mathematics, University of the Basque Country UPV/EHU, 20600 Eibar, Spain; josedavid.nunez@ehu.eus

**Keywords:** fall detection, CNN, LSTM, pruning

## Abstract

Due to the limitations that falls have on humans, early detection of these becomes essential to avoid further damage. In many applications, various technologies are used to acquire accurate information from individuals such as wearable sensors, environmental sensors or cameras, but all of these require high computational resources in many cases, delaying the response of the entire system. The complexity of the models used to process the input data and detect these activities makes them almost impossible to complete on devices with limited resources, which are the ones that could offer an immediate response avoiding unnecessary communications between sensors and centralized computing centers. In this work, we chose to reduce the models to detect falls using images as input data. We proceeded to use image sequences as video frames, using data from two open source datasets, and we applied the Sparse Low Rank Method to reduce certain layers of the Convolutional Neural Networks that were the backbone of the models. Additionally, we chose to replace a convolutional block with Long Short Term Memory to consider the latest updates of these data sequences. The results showed that performance was maintained decently while significantly reducing the parameter size of the resulting models.

## 1. Introduction

In recent years, there has been a continuous increase in the elderly population, both in number and as a percentage of the total population. By 2030, the population over 60 years of age will increase by 0.4 billion and one sixth of them will be over that age. And the trend in the coming years will be similar, reaching 426 million people over 80 years of age by 2050 [1]. Nowadays, there is a clear need for technology that provides efficient and safe assistance to the elderly population to avoid risky situations and help in emergencies. Falls are one of the main causes of deterioration in the quality of life of this population. The origins of these falls can be diverse, and the main risk factors identified in the multivariate analysis were advanced age (>79 years), not having a partner, taking more than two medications, dependence in ADLs (activities of daily living), decreased strength or balance, and walking with technical assistance [2]. Dependence in ADLs can be a consequence of many factors such as reduced flexibility, vision problems, chronic health conditions, and side effects of medications in addition to the risks mentioned above.

A fall detection technology that provides the elderly population with a safe environment and the rapid assistance of healthcare professionals is essential. There are many fall detection systems (FDSs) in the literature that are demonstrated to be efficient in detecting falls and differentiating them from ADLs. However, many of them present many challenges related to the users’ acceptance such as the inconveniences of wearing gadgets that are used in many FDSs and privacy issues. In [3], they also identified challenges regarding performance under real-life conditions, issues related to power consumption, real-time operations, sensing limitations, and records of real-life falls.

The selection of the optimal technology to sense or acquire the information that would alert the caregivers of a possible fall plays a key role in FDSs. The technology used must be tailored to the environment and use case of each application so as not to exceed power and latency requirements. The sensing technology involved in the FDSs should minimize false positives in order to avoid an excessive workload for caregivers. The FDSs can be classified following different strategies, but in this work we divided following the criteria of the selected sensor typology of each work. We divided all the approaches into wearable sensor-based technologies, ambient sensor-based technologies and vision-based technologies. There are some other approaches that use mixtures of the sensor types just mentioned above [4]. The technology that offers the highest reliability is probably the vision-based one, at the expense of complex and resource-consuming computational calculations and long training and inference times. In many applications there are constraints in term of computational resources, latency and memory usage making interesting the use of lightweight models.

In this work we opted for using 2D Convolutional Neural Networks (CNNs) that use as the extra axis the passage of time. Video data are used and divided into sets of frames and these are classified using CNNs in predefined classes. In order to be deployable in the inference process in resource-constrained devices, a pruning technique is applied to reduce the total number of parameters used for representing these networks and preserve the performance of the resultant model. Additionally, a modification of the first model is implemented using a Long Short-Term Memory (LSTM) layer as part of the backbone of the model, adding the capability to detect falls as it is designed to work with sequential data, such as time series data or natural language text [5].

### 1.1. State of the Art

During recent years, important advances have been made in the literature in the human detection and Human Activity Recognition (HAR) field. Different works focused on identifying human activities, and some more specifically on detecting falls. Ultimately, the great majority of the works include artificial intelligence-based techniques to enhance the performance of the entire system. We divided the related works depending on the technology involved in the detection and inference processes, and the data used to identify these activities.

#### 1.1.1. Wearable Sensor-Based Technologies

In many works they opted for using accelerometers and gyroscopes located in strategical positions on the individuals like the waist [6,7], wrist [8], back [9] or shoulders [10] among others. The benefits of this type of system resides in the lack of intrusion strategies to recognize different activities. Another advantage that this type of detection system offers is its lightweight nature, low power consumption and low cost.

In [11], they proposed FallDroid, an automated system for fall detection that offers a prompt response based on smartphone technology using accelerometer data, improving patients’ independence. With the aim of minimizing false positives, they integrated threshold-based methods and multiple kernel learning with Support Vector Machine (SVM) and Multiple Kernel Learning–Support Vector Machine (MKL-SVM) to detect fall-like events. The system’s accuracy, sensitivity and specificity rates show its adequateness to the nature of the paradigm (97.8%, 99.5% and 95.2%). In [12], they proposed an improvement in a threshold-based fall detection system, classifying four type of fall events (forward, backward, left lateral and right lateral falls) and activities of daily life. The patient’s location was immediately transferred as well. Results from the experiments demonstrate its workability, reaching 99.38% accuracy and 96% detection rate.

The addition of other sensing systems to the accelerometers enhances the overall activity recognition. In [13], they developed a FDS adding a portable Inertial Measurement Unit (IMU) to the accelerometers within eyeglasses; in order to identify accidental falls, they employed raw accelerometer and gyroscope data analyzing their signal magnitude and angular head movement data. Additionally, an optimization of angular data was achieved using a complementary filter, accomplished by a threshold-based algorithm. The 95.44% of accuracy achieved distinguishing between falls and non-falls demonstrates the effectiveness of the system in detecting accidental falls, minimizing false alarms and reducing the costs of the employed hardware. In [14], they employed accelerometers and gyroscopes for identifying ADLs and falls. After preprocessing and extracting features from data, various Machine Learning (ML) techniques were adopted for classifying signals from wearable sensors, including SVM, k-Nearest Neighbor (k-NN), Random Forest (RF), and Artificial Neural Networks (ANNs). The model that offered the best performance was SVM achieving 100% recall and 96.34% accuracy in the evaluation process. Not only k-NN offered an impressive performance, but also RF and SVM reached 99% accuracy, sensitivity and specificity.

However, the drawbacks that come with using these sensors make them unsuitable for many environments. Not only are they inconvenient to use, but the high rate of false positives in some scenarios makes them unsuitable for many applications.

#### 1.1.2. Ambient Sensor-Based Technologies

Many researchers opted for using different types of sensors as the source of knowledge for fall detection. RF, sound, IR, pressure and vibration are some types of sensor technologies used in FDSs. The principal benefit that these solutions offer is that they avoid any inconvenience that wearing different gadgets may cause and the lack of intrusion using them.

##### RFID-Based FDSs

Radio Frequency Identification (RFID) technology is used commonly along with other algorithms or sources of knowledge to detect ADLs or falls. This technology uses radio waves for the identification and tracking of different types of objects. The system is based on readers and tags, that are provided with an integrated circuit (IC) that holds data related to the tagged object, accessible from the reader, and an antenna. Depending on the power source employed by these tags, two different types of them can be found. Passive RFID tags work with the energy proceeding from the RFID reader’s radio waves used for transmitting information and activating the IC. On the other hand, active RFID tags have their own power source, that is usually a battery. In [15], they used contactless passive RFID tags for capturing the signal power and phase in an array setup for fall detection. The extraction of human actions was performed in the initial phase using an action segmentation algorithm. A deep residual network then determined falls. The results show its competitive performance with a 96.77% accuracy. In [16], they employed passive RFID tags, employing a wavelet transform for preprocessing the signal data, subsequently applying a SVM to improve the accuracy of the FDS. The system exhibited an accuracy of 96%. However, including more subjects would improve the performance of the system, considering that the extracted parameters were collected from a single subject.

##### Sound-Based FDSs

Sound-based sensors have been implemented in many studies, using audio signals as a source of information about falls. In [17], they proposed a versatile FDS based on an autonomous mobile robot with a built-in microphone. The distinction between falls and non-falls was carried out using sound inputs from bathrooms with an accuracy of 0.8673. In [18], they employed a decision tree for binary and multi-class problems integrating features that consist of melcepstral coefficients, gammatone spectral coefficients and spectral skewness. The results show the method’s reliability for its use in medical centers, nursing homes, old houses and health care provisions, reaching an accuracy of 91.39%, precision of 96.19%, recall of 91.81% and F1-score of 93.95%.

##### IR-Based FDSs

In [19], they proposed a dual-technology sensor (DTS) based on Pyroelectric Infra Red (PIR) and microwaves for motion detection and pressure mats. The experimental results demonstrated the FDS’s adequateness achieving an accuracy, sensitivity and specificity of 89.33%, 100% and 77.14%, respectively. In [20], they incorporated three precise PIR sensors strategically positioned at various points on the wall for a FDS. Fall identification was carried out by considering the alterations in temperature induced by human movement. Additionally, an optical flow methodology was employed to detect and evaluate the motion direction of individuals. The method reached a specificity of 93.7% and a sensitivity of 92.5%. In [21], they employed a Micro-electromechanical Systems Pyroelectric Infrared (MEMS PIR) sensor and a thermopile IR array sensor forming a contactless FDS for detecting bathroom falls. The experimental results demonstrate the feasibility of this FDS, achieving averages of precision, recall, accuracy and F1-score of 94.45%, 90.94%, 92.81% and 92.66%, respectively. Nevertheless, there were some limitations in all these methods in encompassing all possible angles of fall directions, and covering a wide area.

#### 1.1.3. Vision-Based Technologies

The alternative that most studies have adopted for detecting falls and ADLs is to apply vision-based methods. In these approaches, data provided by cameras are used by the detection model to differentiate falls from ADLs. Many of them are followed by a ML-based system to determine the detected activity. In [22], they used depth-map video frames for fall detection, using as input data joint positions obtained by the Kinect sensor. A SVM was used to classify the obtained images. The proposed system achieved an accuracy of 93.6% in the experimental phase. In [23], a vision-based FDS that analyzes body geometry for fall discrimination was introduced, and ML techniques were applied to identify fall patterns. The results demonstrate the system’s high performance achieving 98.32% accuracy, 98.11% precision, 98.11% sensitivity, 98.30% specificity and 98.11% F1-score.

The application of ML-based methods to enhance the predictability of the entire detection system is widely used in the research field. In [24], they detected multiple activities of different individuals in the same scene using as a source of knowledge human skeleton pose estimation for extracting features for activity detection in video camera images. The evaluation of human activities was performed using six ML algorithms, and RF achieved the highest accuracy (95%), evaluated on the UP-Fall [25] dataset. In [26], they implemented diverse ML algorithms to classify test data, including CNN, Logistic Regression (LR), Linear Discriminant Analysis (LDA), K-NN, SVM, Naive Bayes (NB), AdaBoost and RF among others. The experiments showed the capacity of the recognition system for some ML algorithms to identify falls, achieving accuracies of 82%, 85.0%, 39.75%, 66.25%, 68.0%, 64.75%, 72%, 76%, 88% and 94%, for LDA, k-NN, SVM, NB, AdaBoost, RF, Bagging, voting, LR and CNN, respectively.

More sophisticated algorithms are applied ultimately in the field using diverse DL models. In [4], they proposed a model composed of a multi-head CNN with a Convolution Block Attention Module (CBAM) that processes visual data and a Convolutional Long Short-Term Memory (ConvLSTM) network that manages time-sensitive information from various sensors. Evaluating on the UP-Fall dataset, the system achieved an accuracy of 97.44%, F1-score of 97.41%, recall of 97.44% and precision of 97.55%. In [27], a method that combines an autoencoder and three layers of CNN named C3D-AE was proposed. Three-dimensional CNN extracted the features, while the autoencoder modeled typical behaviors. C920 cameras were used for image capturing employing 22 subjects to capture information recording 1760 videos. The system achieved a remarkable 93.3% sensitivity and 92.8% specificity. In [28], they proposed a model composed of an enhanced YOLOv7-X-pose algorithm for a rapid human body keypoint extraction in the multi-person keypoint extraction module, and an enhanced CNN Attention LSTM model, capturing the relevant features of the input sequence and improving the model’s predictability with the addition of the LSTM layer. The extraction algorithm’s extraction order and accuracy were ensured using a Kalman filter target tracking algorithm. The results demonstrated that their application provided reliable support for health management of the elderly. In [29], an enhancement of YOLOv5s was presented and applied as a real-time detection method for identifying falls among the elderly population. The first difference from the original YOLOv5s was the replacement of the existing basic convolution with asymmetric convolution blocks (ACBs) in the convolution module of the backbone network enhancing the feature extraction capability. Additionally, the extraction of feature location was enhanced with the addition of the spatial attention mechanism module to the residual structure of the backbone network. Finally, improving the feature layer structure by removing the feature layer for small targets allows the network to pay more attention to semantic level information. The experimental results demonstrate the improvement from the original YOLOv5s trained and tested on the URFD dataset, with an increase of 3.5% reaching an average accuracy of 97.2%.

In recent years, the irruption of the transformer-based solution has gained special significance in large-scale natural language processing, computer vision, reinforcement learning, audio and robotics among others. In [30], they proposed the first transformer-based solution for a fall detection paradigm tested on the UP-Fall dataset and the UR Fall dataset. The results show that the system achieved a competitive efficiency at differentiating falls and non-falls achieving a 99.17% accuracy for binary classification and a decent performance for multi-class classification with a 93.17% accuracy.

The most prominent solutions in terms of accuracy, deployability and convenience for the elderly population are the wearable sensor-based solutions and vision-based solutions. These last are less susceptible to false positives as they offer more detailed information about the scene of the fall, avoiding extra aids and nursery services. The wearable systems need to be carried by elderly people, becoming an important inconvenience for some of them.

All the aforementioned approaches require long training times, much memory and high capacity processors for the inference phase. However, in certain cases, it is worth considering a solution that omits the usage of centralized solutions and reacts on the edge. In this way, an immediate response is guaranteed, being a critical condition in various applications. For this purpose, it is necessary to reduce the size of the architectures used for predicting these humans’ activities, as well as representing the data with lighter representations. In this work, we opted for using 2D-CNNs for extracting features of video frames corresponding to falls. We omitted the possibility of using transformer-based solutions due to their complexity and lack of knowledge at reducing their models. Later, we propose an alternative adding a LSTM block to enhance the predictability of the model giving more capability to handle time-sensitive information to the model. Finally, we apply well-known reduction techniques to two of the layers of the backbone of our proposed model to make it deployable on the edge.

## 2. Materials and Methods

This section describes the materials used in the experimental process and the methodology followed to detect falls in the elderly population, as well as the procedure for the evaluation process of the applied models using the datasets mentioned in this section.

### 2.1. Feature Extraction Models

In this work, we opted for using CNNs as the backbone model for detecting the elderly population’s falls. These are NNs composed of convolution layers that are appropriate at extracting features from images and video frames. In this case, we intended to use the sequences of falls to train the models and determine whether a certain motion of a person was going to end as a fall or not. For this purpose, we added an extra dimension to these convolutional networks adding the time feature to the data by concatenating consecutive image frames and using them as input to the 2-dimensional CNNs. We concatenated 3 convolutional blocks in the first proposed model and in the second proposed model we substituted the last convolutional block with the LSTM block. Figure 1 gives a graphical representation for both models.

The backbone of the first model was compounded of 3 consecutive convolutional blocks and 3 consecutive dense layers. Each of the convolutional blocks was composed of a 2-dimensional convolution layer and a MaxPooling layer for summarizing the features of a region of the feature map generated by the convolution layer adding a small amount of translation invariance. Following, a batch normalization layer was used to re-center and re-scale the inputs for the posterior layers for enhancing the training of the models by accelerating and making them more stable. Finally, the dense layers were concatenated each of them followed by a dropout layer for avoiding overfitting issues. Table 1 specifies the details of the model, and the model diagram is given in Figure 1.

The second model used in the experiments differed from the first one principally in the third block of the backbone. In the first model, the third block was made up of a convolution layer and a MaxPooling layer, but in this case the third block was made up of a LSTM layer. This gives to the model ability to consider time-sensitive features of the input data. At the same time, reduction in the number of parameters needed to represent the entire model is favorable due to the difference between the parameters needed to represent a LSTM layer and a convolutional block. Finally, to match with the input data type of the LSTM layer, we applied 1D convolutional layers, and reshaped the input data as sequences of images binding them. Table 2 summarizes the specifications of the second model.

### 2.2. Reduction Techniques: Sparse Low Rank Method (SLR)

As our goal was to deploy these models on devices with reduced memory sizes and tiny processors, we performed pruning techniques to remove redundant elements of certain layers of each model.

We applied the Sparse Low Rank Method (SLR) proposed in [31]. In this case, singular value decomposition (SVD) was applied to the weight matrix of a given layer, decomposing the original matrix into rotation, rescaling and flip matrices. These resulting matrices required far fewer parameters to represent, which is beneficial for use on resource-constrained devices. The activation $a\in R^{n}$ of a Fully Connected (FC) layer with m input and n output neurons is represented as
(1)$$a=g(W^{T}X+b)$$
where X represents the input data and **g**() represents the activation function. Each parameter in the weight matrix W is $w^{ij}\in R$(1≤i≤m,1≤j≤n), and bias matrix b is $b^{j}\in R\left(1\leq j\leq n\right)$. The proposed approach was applied to the weight matrix W after adjusting its weights in the training process. In addition, SVD disintegrated the weight matrix W as $W=USV^{T}$ where $U\in R^{m\times m}$ and $V^{T}\in R^{n\times n}$ are orthogonal matrices and $S\in R^{m\times n}$ is a diagonal matrix, where the components with the highest absolute value are located in the first rows and columns of the U and $V^{T}$ matrices. Among the different alternatives to consider the most relevant rows and columns of these matrices we opted for using the rows and columns whose weights’ absolute values were the highest. In one of our previous works where we applied this reduction technique [32], we concluded that in terms of time efficiency it was much more feasible to apply the weights’ absolute value as the criteria for the most relevant components selection. Although the cost defined in [31] could give a slight improvement in terms of accuracy of the reduced models, the severe increment of the training time does not compensate for this accuracy improvement.

Moreover, the SLR method only considers the most relevant rows and columns of each of the rows and columns of the rotation matrices, even achieving a higher compression of the original matrix. Applying the reduced rank (rk), only the first k rows of U and columns of ${\hat{V}}^{T}$ are kept, based on the weight criteria.

Pruning FC layers is much more effective in terms of accuracy, time and energy efficiency than pruning convolution layers as shown in [33]. Thus, we applied this reduction technique to the first 2 FC layers of each of our backbone models, leaving the last FC layer untouched in each case.

### 2.3. Datasets

The data used to determine the usefulness of the aforementioned models to detect falls with a significant reduction in the models’ size, while preserving the predictability of the resultant models, are summarized in this section. Open well-known public datasets were used in the training and validation processes: UP-Fall dataset [25] and Multiple Cameras Fall dataset [34].

UP-Fall Detection dataset is composed of raw and feature sets retrieved from 17 healthy young individuals without detriments that executed 3 attempts for 11 different activities and falls. Between the eleven differentiable activities, six of them were simple human daily activities and five different types of human falls. The collection was completed in a period of four weeks, from 18 June to 13 July 2018 on the third floor of the Faculty of Engineering, Universidad Panamericana, Mexico City, Mexico. All the devices and equipment for measurements were connected locally to a set of computers which centralized all the information and saved the data on hard drives.

They performed the entire recording process in a controlled laboratory room without variations in light intensity, with all the cameras remaining in the same position during the process. They considered three sources of knowledge for a multimodal approach, using wearables, context-aware sensors and cameras concurrently. However, in our work we used only the data obtained by the cameras. See Figure 2.

The activities performed by the subjects can be classified into two main divisions:Falls: falling forward using hands, falling forward using knees, falling backwards, falling backwards and falling sitting in empty chair.Non-falls: walking, standing, sitting, picking up an object, jumping and lying down.

Multiple Cameras Fall dataset is a video dataset that contains simulated falls and normal daily activities acquired in realistic situations. See Figure 3. The multi-camera system used in the recording phase is composed of eight inexpensive IP cameras with a wide angle to cover all the recording room. The dataset is composed of several simulated normal daily activities and falls viewed from all the cameras and perform by one subject. All the activities recorded can be classified into two main divisions:Activities of daily living that include walking in different directions, housekeeping, activities mistakable for falls (sitting down/standing up, crouching down) and image processing difficulties like occlusions or moving objects.Different types of simulated falls: forward falls, backward falls, falls when inappropriately sitting down, loss of balance. These were completed in different directions with respect to the camera.

## 3. Experimental Process

In this section, details of the training and evaluation process of the models and a summary of the results obtained when the models were tested on the UP-Fall dataset and the Multiple Camera Fall dataset are given.

### 3.1. Training and Validation

For the evaluation of the proposed methodologies we tested our two alternatives of the CNN models on the datasets described above, and we compared against relevant works from the literature. We compared different performance metrics, the parameter size used for representing the entire model and the time needed to train each model using each dataset.

For training and validating the models, we followed a 10-fold Cross-Validation strategy, that includes resampling and sample splitting methods that use different portions of the data to test and train a model on different iterations. In this way, we guarantee that the results are independent of the partition between the training and test data. We regarded the accuracy, precision and recall in each case, these being micro-averaging metrics. The models’ hyperparameters were adjusted manually after a first intuitive deduction of best configurations. Consequently, the parameters used in the training process were the following. The optimizer was Adadelta (learning rate = 0.98, loss = categorical cross entropy), batch size was 32 and the number of epochs was 50. In the first dataset case, differences between reducing different FC layers and these reductions’ sizes were compared to understand the effect of these reductions and achieve the optimal reduction rank that offers the best relation between reduction and performance. In both cases, FC6 and FC7 were the selected layers for the pruning process on account of the fact mentioned in Section 2.2.

As our intention was to reduce the size of the models to fit into resource-constrained devices in the inference model while keeping the prediction capacity almost intact, we looked at the reduction in the resultant model size after applying a pruning technique to each model to consider only the most relevant neurons of certain layers of each model. As mentioned above, this reduction technique was developed in the first two FC layers of the backbone of each model, and compared with different works from the literature that test their proposed method in the UP-Fall dataset. We applied different reduction ranks so as to achieve a significant reduction in the memory size used to represent the resultant networks, and we present the ones that offer the best relation between performance and parameter reduction. The parameter number and the memory size needed for each model were analyzed along with the performance metrics mentioned above.

### 3.2. Software and Hardware Specifications

All the experimentation process was developed using the Python 3.11.7 programming language, using different libraries to deal with ML tasks such as Tensorflow 2.15.0, Keras 2.15.0, Scikit-Learn 1.4.1.post1 and others to deal with data analysis like Numpy 1.26.4 or Pandas 2.2.1.

The hardware specification on which the experiments took place are the following:Dell Precision 7560. Round Rock, TX, USA.Intel i7-11850H working at 2.5 GHz. Santa Clara, CA, USA.32 GB DDR4 RAM;x64 Windows 10 Professional operating system. Redmond, WA, USA.

### 3.3. Experimental Results on the UP-Fall Dataset

This section collects the performance results obtained when each of the models were trained and reduced on the UP-Fall dataset, and compared with various alternatives from the literature. To deduce the best compression rate for the FC layers that were pruned, we tested different reduction ranks and compared their performance metrics. In principle, the performance metrics should be higher for the cases where the most components from the original sparsified matrices were kept, but that was not always the case. In some cases, part of the components of certain layers are redundant and worsen the learning process of the entire model. In these cases, the elimination of these redundant components of the layers may improve the performance of the entire model. Table 3 summarizes the comparison of different reductions. The reduction rank1 and rank2 refer to the number of rows and columns considered in the sparsified matrices of the first and the second FC layer sparsified from the original backbone models. Additionally, reduction rank (0.5) was applied to consider only half of the elements of each row and column.

There is a totally different trend for the reduction variation for each model. In the case of the model consisting of only convolutional blocks when more elements are pruned from the original weight matrices, the results are worse. This is due to the loss of information between the connection of neurons from the subsequent layers. However, in the case of the model consisting of convolutional blocks and LSTM, the best results were obtained when the reduction rank = 8 was applied, as it can be observed in Table 4. This means that, when more components are considered, these hinder the learning process. The elimination of these redundant components improves the final classification result and lightens the model. We tested reducing this reduction parameter even more, but when the rank was even lower the results deteriorate.

To compare the results of each model with a given reduction rate and the results obtained applying the model proposed in [35], we performed a comparative study on the UP-Fall dataset. The performance metrics for rank = 8 are given along with the parameter size of each model and a comparison of their training times in Table 5.

In order to visualize the effect of pruning in our models we made a comparative graph that shows the difference in accuracy between our two models in their original version and their pruned versions. Figure 4 shows this comparative analysis.

Finally, we made a comparison with several results obtained in various works that tested their proposed technologies on the UP-Fall dataset. Table 6 summarizes the accuracies obtained in the evaluation process of various works when tested on the UP-Fall dataset as a multi-class problem.

### 3.4. Experimental Results on the Multiple Cameras Fall Dataset

To support the results obtained in the previous dataset, the two models and the alternative from [35] were tested on the Multiple Fall dataset, applying the optimal reduction ranks for this case, obtained in an exhaustive search. The performance metrics obtained when each of the models were trained and reduced on the Multiple Fall dataset are given within this subsection.

As it can be observed in Table 7, the reduction in parameter size is considerable between the model that was designed in [35] and the reduced models we achieved applying the SLR method. There is a non-negligible performance drop when the FC layers are pruned. However, a satisfactory performance is achieved by both models after reducing their FC layers. In comparison to the previous dataset, in this case, the performance drop is less drastic, being a less than 4% drop for the 2D-CNN and SLR model.

## 4. Discussion

The above section summarizes the performance variations when SLR pruning methodology was applied to the models proposed in Section 2 and the comparison with various models from the literature as well. The reduction of the models generally involves a loss of information between layers of the backbone model, but in some scenarios the redundancy of some elements carries a downgrade of the model, the elimination of these elements being precise. Although this phenomena was perceptible in the model composed of LSTM and convolutional blocks, it did not result in the same way in the model composed of convolutional blocks solely. The best performance metrics were achieved in the case when the reduction rank reached the value of 8 in the 2D-CNN-LSTM model, whereas the best metrics were achieved when the reduction was lighter in the case of the 2D-CNN model, apart from the cases where the entire model was used for classifying scenes. In both cases, the parameter reduction achieved when the weight matrix of a certain layer was represented by the sparsified matrices and these were pruned based on the SLR methodology was notorious and may be a key factor when embedding these fall detection models in resource-constrained devices.

The selection of the optimal layer to reduce is critical in order to maintain the resultant model’s accuracy as high as possible and to achieve a significant reduction in the parameter size and consequent use of memory in the embedded device. In this sense, the convolutional layers have a great capability for extracting features from input data and these should be preserved in order to maintain the model’s performance. Between the FC layers, it is worth noting that the removal of weights from the layers that are closer to the convolutional blocks carried a little more reduction in the resultant model’s accuracy, a better alternative being the application of more aggressive pruning in the FC layers that are close to the convolutional blocks.

Compared with the results obtained in various works from the literature, our reduced models cannot approximate to the performance achieved by various models from the literature. In contrast, our models achieved a higher performance when no reduction was applied to any of the layers that compose the backbone model. If a reduction in parameter size was required in order to fit these models on resource-constrained devices, a pruning of these models was required to not exceed the memory restriction that would be found in embedded devices. For that purpose, a precise pruning technique guarantees a less drastic reduction in the predictability of the resultant model. Accordingly, the reduced models proposed in this work maintained a good performance even when the reduction in parameters achieved values close to 94%.

Regarding the different percentage of parameter reduction in the pruned models, the best results were to be expected when this percentage was lower, i.e., when the majority of the components were conserved. In spite of the incongruity, when the reduction was harder when applied to the 2D-CNN-LSTM model, it classified better than when more components were conserved from the original model. The reason why this unusual case occurred is that part of the components that were eliminated from the original matrices were redundant and worsened the predictability of the model. Although some components were better to be pruned, in this pruning process key elements for correct classification may be removed as well. Thus, the original model’s performance metrics were superior to the ones obtained by the reduced model in all cases.

Between the proposed models, the variations in the performance metrics when the entire model was applied in the inference process were negligible. However, when the reduction techniques were applied to lighten the models to enable their implementation on resource-constrained devices, the 2D-CNN-LSTM model was superior when tested on the UP-Fall dataset. The inclusion of the LSTM enhanced the classification of falls due to the ability of LSTM blocks to predict future events and signals.

In fact, when these models were applied on the UP-Fall dataset, their performance was superior to the majority of the solutions that could be found in the literature. In cases where the accuracy needs to be outstanding and excess of memory usage is not a drawback, the application of any of the backbone models proposed in this article would be interesting. In contrast, for the majority of the embedded devices, the memory restrictions do not allow their application in these type of devices. To this end, the application of an appropriate reduction technique becomes essential, where SLR showed excellent results with good reductions in parameter sizes and decent accuracy of the reduced models.

## 5. Conclusions and Future Work

In this article, we have demonstrated that when applying appropriate reduction techniques to models whose backbone is composed of CNN and models that combine CNN and LSTM the resultant model is able to maintain a decent accuracy while reducing the parameter size by more than %90. The SLR method was applied to two of the backbone models proposed in Section 2, and different reduction rates were applied to two of the FC layers of these backbone models to tackle the most appropriate reduction rate. These reduced models and the one proposed by [35] were trained and tested on the UP-Fall dataset and Multiple Camera Fall dataset, and many other works from the literature were compared on the UP-Fall dataset. Obviously, the non-pruned models offered a better prediction ability than the pruned models, but these needed much more memory to store their parameters and their implementation became complicated in many resource-constrained devices. However, the reduced models achieved reductions up to 93% and their accuracy reached values close to the complete models’ versions. Moreover, the backbone models without applying any pruning techniques were superior to the model compared from the literature.

To add more robustness to the proposed models, it would be of particular interest to add an uncontrolled environment where the subjects would be the elderly population. In these cases, the randomness of the subjects’ decisions due to different factors such as dementia or health problems would complicate the correct functioning of the resulting model. This would imply a higher degree of uncertainty in the experimentation and the system in general should offer better reproducibility when used in real-world applications. In general, the validation results would not meet the performance metrics achieved on datasets collected in controlled environments, but would mimic the functionality of real-world applications.

As future work, the deployment of the pruned models on actual embedding devices would strengthen the suitability and adequateness of the models for this paradigm. In addition, the development of a novel technology to reduce the models employed in this work may better suit this paradigm. A novel reduction technique would consider the characteristics of the final device where the model would be deployed, the features of the original model employed for inference and the characteristics of the input dataset. In this way, an optimization could be reached even though our method has achieved a remarkable reduction in model size at the expense of a slight reduction in performance.

Additionally, the comparison of these methodologies in another testing environment would be interesting, apart from the fall prediction paradigm. These reduction techniques would be suitable for more types of models, including those used in gas leaks, for example. In these scenarios, embedded devices equipped with tiny sensors and processors would benefit from these reduced models to successfully detect gas leaks. 

## Figures and Tables

**Figure 1 sensors-24-07256-f001:**
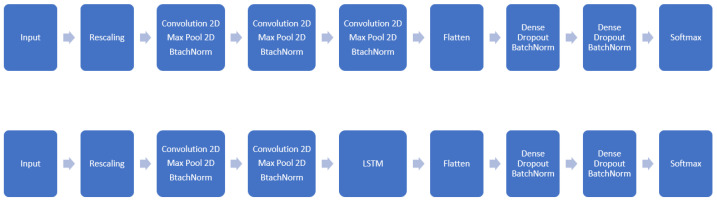
Proposed backbone models’ diagram.

**Figure 2 sensors-24-07256-f002:**
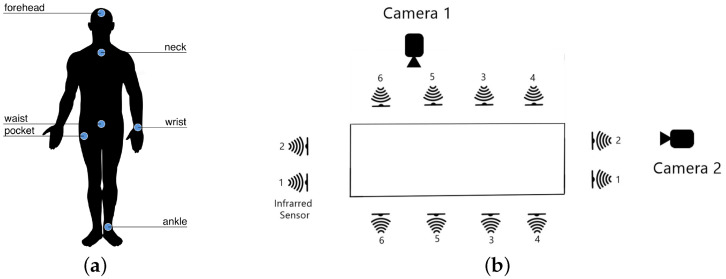
UP-Fall Dataset recording setup. (**a**): Location of motion sensors. (**b**): Location of cameras. Source: [25].

**Figure 3 sensors-24-07256-f003:**
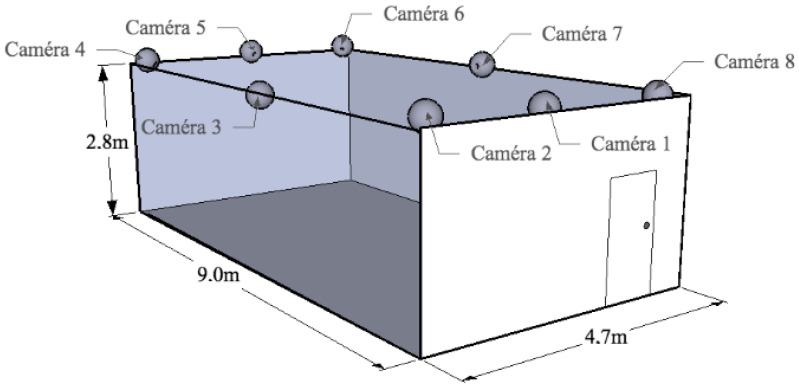
Multiple Dataset recording setup. Source: [34].

**Figure 4 sensors-24-07256-f004:**
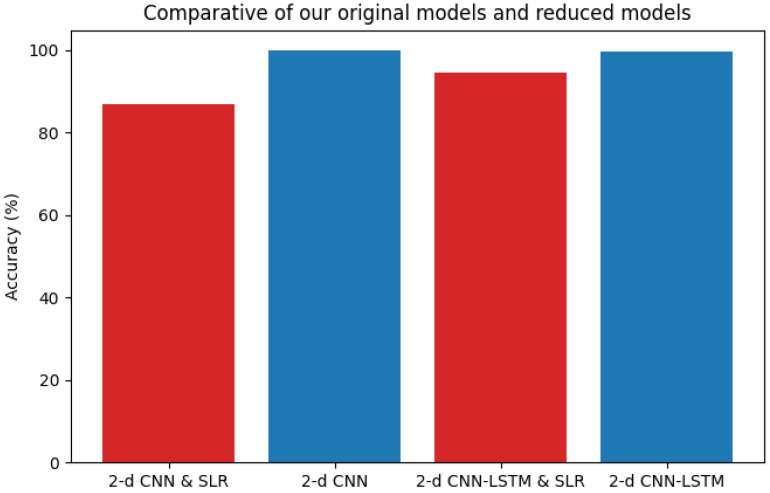
Comparative graph of pruned vs original versions.

**Table 1 sensors-24-07256-t001:** First model. Two-dimensional CNN model.

Layer Name	Layer Type	Feature Map	Output Size of Images	Kernel Size	Stride	Activation
Input	-	1	36 × 32 × 32 × 3	-	-	-
Conv-1	2D Conv	8	36 × 8	3 × 3	1	relu
Pool1	Maxpool	8	36 × 8	3 × 3	2	relu
Conv-2	2D Conv	32	36 × 32	3 × 3	1	relu
Pool2	Maxpool	46	36 × 32	3 × 3	2	relu
Conv-3	2D Conv	128	36 × 128	3 × 3	1	relu
Flatten	Flatten	-	1472	-	-	relu
FC6	Dense	-	1200	-	-	relu
FC7	Dense	-	600	-	-	relu
FC8	Dense	-	# of classes	-	-	softmax

**Table 2 sensors-24-07256-t002:** Second model. CNN-LSTM model.

Layer Name	Layer Type	Feature Map	Output Size of Images	Kernel Size	Stride	Activation
Input	-	1	192 × 1024 × 3	-	-	-
Conv-1	1D Conv	8	190 × 8	3 × 3	1	relu
Pool1	Maxpool	8	95 × 8	3 × 3	2	relu
Conv-2	1D Conv	32	93 × 32	3 × 3	1	relu
Pool2	Maxpool	46	46 × 32	3 × 3	2	relu
LSTM	LSTM	-	46 × 32	-	-	relu
Flatten	Flatten	-	1472	-	-	relu
FC6	Dense	-	1200	-	-	relu
FC7	Dense	-	600	-	-	relu
FC8	Dense	-	# of classes	-	-	softmax

**Table 3 sensors-24-07256-t003:** Performance metrics of classification of UP-Fall dataset for different reductions applied on 2D-CNN and SLR models.

Reduction Rank	8	12	16	No Reduction
Accuracy (%)	86.82	92.76	95.29	99.93
Recall (%)	88.36	94.69	96.35	99.88
Precision (%)	86.66	92.41	93.71	99.39
Parameter reduction (%)	92.34	91.59	90.79	0

**Table 4 sensors-24-07256-t004:** Performance metrics of classification of UP-Fall dataset for different reductions applied on 2D-CNN-LSTM and SLR models.

Reduction Rank	8	12	16	No Reduction
Accuracy (%)	94.47	91.73	78.71	99.82
Recall (%)	94.40	91.69	79.24	99.85
Precision (%)	93.31	90.65	77.64	99.37
Parameter reduction (%)	93.79	92.54	91.28	0

**Table 5 sensors-24-07256-t005:** Performance metrics of classification of UP-Fall dataset for different models.

Model	2D-CNN and SLR	2D-CNN-LSTM and SLR	[35]
Accuracy (%)	86.82	94.47	99.39
Recall (%)	88.36	94.40	99.39
Precision (%)	86.66	93.31	99.40
Parameter size (MB)	1.96 ^1^	0.90 ^2^	177.90

^1^ Reduction rank1 = 8; reduction rank2 = 8. ^2^ Reduction rank1 = 8; reduction rank2 = 8.

**Table 6 sensors-24-07256-t006:** Accuracies from various works from SoA tested on UP-Fall dataset.

Work	Input	Accuracy (%)
Martínez-Villaseñor et al. (2019) [25]	Sensor	95.49
Ramirez et al. (2021) [36]	Skeleton	99.45
Ha et al. (2022) [35]	RGB + Sensor	99.39
Moha Gouda et al. (2022) [37]	RGB + Sensor	99.2
Islam et al. (2023) [4]	RGB + Sensor	97.90
Yan et al. (2023) [38]	Skeleton + Sensor	98.05
Nuñez-Marcos et al. (2024) [30]	RGB	93.17
Ours (2D-CNN)	RGB	99.93
Ours (2D-CNN-LSTM)	RGB	99.82
Ours with pruning (2D-CNN and SLR)	RGB	86.82
Ours with pruning (2D-CNN-LSTM and SLR)	RGB	94.47

**Table 7 sensors-24-07256-t007:** Performance metrics of classification of Multiple Fall dataset for different models.

Model	2D-CNN and SLR	2D-CNN-LSTM and SLR	[35]
Accuracy (%)	95.45	92.16	99.09
Recall (%)	95.27	92.00	98.79
Precision (%)	95.61	92.29	99.17
Parameter size (MB)	1.96 ^1^	0.90 ^2^	177.90

^1^ Reduction rank1 = 64; Reduction rank2 = 16. ^2^ Reduction rank1 = 64; Reduction rank2 = 8.

## Data Availability

All data used in the experimental process are available online, the Multiple Cameras Fall dataset at https://www.iro.umontreal.ca/~labimage/Dataset/ ( accessed on 27 June 2024) and the UP-Fall dataset at http://sites.google.com/up.edu.mx/har-up/ ( accessed on 9 April 2024).

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
