# Peer review of "Reduction of Vision-Based Models for Fall Detection"

_sensors, 2024, doi:10.3390/s24227256_

Round 1
Reviewer 1 Report
Comments and Suggestions for Authors
The paper presents a method for optimizing vision-based fall detection models by applying pruning techniques (Sparse Low Rank Method, SLR) to reduce the computational and memory requirements of Convolutional Neural Network (CNN) and CNN-LSTM architectures. The authors aim to create efficient models deployable on resource-constrained devices while maintaining performance accuracy. The models are evaluated using two publicly available datasets, UP-Fall Detection and Multiple Cameras Fall Dataset.
-
Relevance and Timeliness:
- The paper addresses an important problem—fall detection for elderly populations, which is increasingly relevant due to global aging demographics. Developing lightweight models that can be deployed on edge devices is a valuable contribution in the field of Internet of Things (IoT) and healthcare technologies.
-
Methodological Innovation:
- The application of the Sparse Low Rank (SLR) Method for pruning CNN and CNN-LSTM architectures is interesting. The use of these pruning techniques to optimize models specifically for resource-constrained environments shows innovation and practical application potential.
-
Experimental Validation:
- The paper employs two well-known datasets (UP-Fall and Multiple Cameras Fall Dataset) to evaluate the performance of their models. The use of multiple datasets adds credibility and robustness to the findings.
-
Comprehensive Analysis:
- The authors provide a detailed comparison of model performance before and after pruning. They analyze various reduction ranks, demonstrating a good understanding of the impact of parameter reduction on model accuracy and computational efficiency.
-
Clarity and Readability:
- The language used in the paper has several grammatical errors and unclear phrasing, which affect its readability. The paper would benefit from professional proofreading to ensure clarity and precision in technical explanations.
-
Lack of Novelty in the Baseline Approach:
- While the pruning technique is well-explained, the underlying models (CNN and CNN-LSTM) are standard. It might be useful for the authors to justify why these specific architectures were chosen or to compare them with more advanced models in the literature, such as Transformer-based architectures, which are gaining traction in vision tasks.
-
Limited Scope of Evaluation:
- The experiments are conducted using two datasets, which are both controlled and structured. Real-world testing would significantly enhance the validity of the results. The paper could include a discussion on the performance gap between controlled settings and real-world applications or suggest a future plan to address this limitation.
-
Shallow Discussion of Results:
- The discussion section focuses primarily on numerical results without delving deeply into why certain reductions perform better or worse. A deeper analysis of the implications of specific parameter reductions and their relationship to the model's interpretability and robustness would add value to the work.
-
Insufficient Comparison with State-of-the-Art:
- The paper compares its approach with only one other model (reference [35]). Expanding the comparison to include a broader set of recent fall detection systems could provide a more comprehensive evaluation of the proposed method’s strengths and weaknesses relative to existing solutions.
-
Scalability and Real-World Application Feasibility:
- Although the authors claim that the reduced models are suitable for deployment on resource-constrained devices, the paper does not provide specific hardware or real-world testing scenarios to validate this claim. Demonstrating the models on actual embedded devices would strengthen the argument.
Recommendations for Improvement
-
Enhance Clarity and Language:
- Professional editing and proofreading should be employed to improve the clarity and readability of the manuscript. This will help convey the technical details more effectively.
-
Broaden the Experimental Scope:
- Incorporate real-world tests or at least provide a detailed discussion of the potential differences between the controlled dataset environment and real-world deployment. This could involve simulated tests with more diverse environmental conditions and subject variability.
-
Expand Comparison with State-of-the-Art:
- The paper should include a broader review of state-of-the-art fall detection models and compare the performance of the proposed models with multiple contemporary methods to showcase its relative advantages.
-
Deepen Analysis of Results:
- Expand the discussion section to explain why certain parameter reductions yield better results and explore the implications of this for model generalizability and robustness.
-
Validation on Real Embedded Devices:
- Demonstrate the deployment of the pruned models on actual embedded devices (e.g., Raspberry Pi or similar hardware) to provide evidence of their suitability for edge applications.
- The language used in the paper has several grammatical errors and unclear phrasing, which affect its readability. The paper would benefit from professional proofreading to ensure clarity and precision in technical explanations.
Author Response
Comments 1. Clarity and Readability.
Response 1. Several changes have been made to improve the readibility of the manuscript, correcting grammar and phrasing mistakes.
Comments 2. Lack of Novelty in the Baseline Approach.
Response 2. The latest models from the literature have been included in the last paragraphs of the SoA section and the election of the proposed model has been justified.
Comments 3. Limited Scope of Evaluation:.
Response 3. In the conclusion section the inclusion of a non-controlled scenario in the training and validation process is considered, citing the benefits that would offer testing in such conditions.
Comments 4. Shallow Discussion of Results.
Response 4. As the components that have been removed are not identified the impact that these should have in the resultant model is not discussed. However, the impact of the election of the layer to be pruned is justified in the discussion.
Comments 5. Insufficient Comparison with State-of-the-Art.
Response 5. The evaluation process was made comparing with a broad set of works from the literature and the discussion was made considering them.
Comments 6. Scalability and Real-World Application Feasibility.
Response 6. In the conclusion section we stated as future work the implementation of our proposed models in a real-world scenario.
Reviewer 2 Report
Comments and Suggestions for Authors
The paper chooses to use a 2D convolutional neural network (CNN). Video data is segmented into a series of frames, which are classified into predefined categories using CNN. To enable deployment in inference processes on resource-constrained devices, a pruning technique is applied to reduce the total number of parameters used to represent these networks while maintaining good performance. Furthermore, a modification is made to the first model by adding a Long Short-Term Memory (LSTM) layer to the backbone, which enhances the ability to detect falls. However, there are still some issues that need to be addressed.
1. Please elaborate on why a vision-based solution was chosen over wearable sensor technology or environmental sensor technology, and highlight the contributions of this method.
2. The chapter arrangement is unreasonable. For example, section 2.5 should be placed in the results chapter; section 2.2 does not need to be divided into 2.2.1; the title of section 3.1 should be “Experimental Results on the UP-fall Dataset,” and the title of section 3.2 should be “Experimental Results on the Multiple Cameras Fall Dataset,” etc.
3. The paper lacks sufficient originality. Although the authors apply SLR technology to optimize the CNN model and attempt to replace certain modules in the CNN with LSTM, this approach does not demonstrate significant innovation or improvement. The current work resembles a simple application of existing technology, lacking a new theoretical framework or unique experimental design. It is recommended that the authors consider more innovative methodologies to enhance the academic value of the paper.
4. There are formatting issues in the paper. For example, lines 345-147 should be described using a table or a paragraph; the formatting of lines 113, 132, and 144 needs to be modified; the chapter naming also needs revision; section 2.1 “Applied Models” should be more specific; figures 1 and 2 are contents from published papers and should be properly cited in the references.
5. In section 2.1, a clearer explanation of the proposed method is needed, and an overall framework diagram of the model can be provided for a more intuitive understanding of the model.
6. Add performance metrics such as the number of parameters, model size, FPS, etc., to increase the rigor of the evaluation process.
7. When applying the SLR pruning technique to the model, a more detailed validation of SLR's effectiveness is required; please provide visual results comparing the models with and without SLR.
8. The training and validation process needs a more detailed description, such as the experimental workflow and the setting of model hyperparameters, so that readers can better study this method.
9. Please include performance metrics for the unpruned CNN and CNN-LSTM models in the evaluation phase to illustrate that “the backbone model without any pruning technique outperforms the models compared in the literature.”
Comments on the Quality of English Languagesee Comments and Suggestions for Authors
Author Response
Comments 1. Please elaborate on why a vision-based solution was chosen over wearable sensor technology or environmental sensor technology, and highlight the contributions of this method.
Response 1. An explanation of the reason why the selected technology was used in this paradigm is given at the end of section 1.
Comments 2. The chapter arrangement is unreasonable. For example, section 2.5 should be placed in the results chapter; section 2.2 does not need to be divided into 2.2.1; the title of section 3.1 should be “Experimental Results on the UP-fall Dataset,” and the title of section 3.2 should be “Experimental Results on the Multiple Cameras Fall Dataset,” etc.
Response 2. This has been fixed with the rearrangement of the sections with the suggested modifications.
Comments 3. The paper lacks sufficient originality. Although the authors apply SLR technology to optimize the CNN model and attempt to replace certain modules in the CNN with LSTM, this approach does not demonstrate significant innovation or improvement. The current work resembles a simple application of existing technology, lacking a new theoretical framework or unique experimental design. It is recommended that the authors consider more innovative methodologies to enhance the academic value of the paper.
Response 3. We included at the end of Section 1 an explanatory of the selection of the actual models, including possibilities of the latest variants from the literature.
Comments 4. There are formatting issues in the paper. For example, lines 345-147 should be described using a table or a paragraph; the formatting of lines 113, 132, and 144 needs to be modified; the chapter naming also needs revision; section 2.1 “Applied Models” should be more specific; figures 1 and 2 are contents from published papers and should be properly cited in the references.
Response 4. Suggested formatting issues have been fixed.
Comments 5. In section 2.1, a clearer explanation of the proposed method is needed, and an overall framework diagram of the model can be provided for a more intuitive understanding of the model.
Response 5. More details about the models are given complemented with a graphical representation of them.
Comments 6. Add performance metrics such as the number of parameters, model size, FPS, etc., to increase the rigor of the evaluation process.
Response 6. Parameter size is given in Tables 5 and 6 along with the performance metrics of all the variants tested in both datasets.
Comments 7. When applying the SLR pruning technique to the model, a more detailed validation of SLR's effectiveness is required; please provide visual results comparing the models with and without SLR.
Response 7. The validity of the SLR method is proved in the results section with the significant reduction of the parameter size along with conservation of the resultant model's predictive capability.
Comments 8. The training and validation process needs a more detailed description, such as the experimental workflow and the setting of model hyperparameters, so that readers can better study this method.
Response 8. The entire experimental process section has been reorganized giving more details of the training and validation process and the setting of the model hyperparameters.
Comments 9. Please include performance metrics for the unpruned CNN and CNN-LSTM models in the evaluation phase to illustrate that “the backbone model without any pruning technique outperforms the models compared in the literature.”
Response 9. Tables 3 and 4 include these metrics under the column "No reduction".
Round 2
Reviewer 1 Report
Comments and Suggestions for Authors
The manuscript is improved, and it can be accepted.
Author Response
We have made some minor changes to respond the comments of the Reviewer 2.
Reviewer 2 Report
Comments and Suggestions for Authors
Dear Editor,
I have carefully reviewed the authors' revisions and additions and would like to provide the following feedback:
1.The formatting issues in the article have been largely addressed; however, line 270 could be deleted or replaced with the title of section 2.2 . I also advise the authors to carefully check the remaining part of the manuscript.
2.Regarding the validation of SLR effectiveness, the authors have provided necessary comparisons in the results section. However, I suggest that they include a comparative graph to visualize the effects of applying the SLR technique versus not applying it. This would enhance the understanding of the technique's impact.
3.The authors have responded to my comments thoughtfully and revised the paper effectively. While the quality of the paper has improved, it still requires greater depth and innovation in terms of originality. I encourage the authors to consider these suggestions in their future research to further enhance the academic contribution of the paper.
see Comments and Suggestions for Authors
Author Response
Comments 1: The formatting issues in the article have been largely addressed; however, line 270 could be deleted or replaced with the title of section 2.2 . I also advise the authors to carefully check the remaining part of the manuscript.
Response 1: Line 270 has been deleted and title of section 2.2 has been modified. The rest if the manuscript has been checked.
Comments 2: Regarding the validation of SLR effectiveness, the authors have provided necessary comparisons in the results section. However, I suggest that they include a comparative graph to visualize the effects of applying the SLR technique versus not applying it. This would enhance the understanding of the technique's impact.
Response 2: We included a graph that shows the difference in accuracy of our methods in their original version and pruned.
Comments 3: The authors have responded to my comments thoughtfully and revised the paper effectively. While the quality of the paper has improved, it still requires greater depth and innovation in terms of originality. I encourage the authors to consider these suggestions in their future research to further enhance the academic contribution of the paper.
Response 3: We expand the paragraph corresponding to future work by clarifying the extra improvement that the implementation of a novel algorithm would entail to reduce the model used in this work.